# PROPER NETWORK INTERPRETABILITY HELPS ADVERSARIAL ROBUSTNESS IN CLASSIFICATION

## ABSTRACT

Recent works have empirically shown that there exist adversarial examples that can be hidden from neural network interpretability **(namely, making network interpretation maps visually similar)**, and interpretability is itself susceptible to adversarial attacks. In this paper, we theoretically show that with the correct measurement of interpretation, it is actually *difficult* to hide adversarial examples, as confirmed by experiments on MNIST, CIFAR-10 and Restricted ImageNet. Spurred by that, we develop a novel defensive scheme built only on robust interpretation (without resorting to adversarial loss minimization). We show that our defense achieves high classification robustness, outperforming state-of-the-art adversarial training methods against attacks of large perturbation while attaining high interpretation robustness under various settings of adversarial attacks.

## 1 INTRODUCTION

It has become widely known that convolutional neural networks (CNNs) are vulnerable to *adversarial examples*, namely, perturbed inputs with intention to mislead networks' prediction (Szegedy et al., 2014; Goodfellow et al., 2015; Papernot et al., 2016a; Carlini & Wagner, 2017; Chen et al., 2018; Su et al., 2018). The vulnerability of CNNs has spurred extensive research on adversarial attack and defense. To design adversarial attacks, most work has focused on creating either imperceptible input perturbations (Goodfellow et al., 2015; Papernot et al., 2016a; Carlini & Wagner, 2017; Chen et al., 2018) or adversarial patches robust to the physical environment (Eykholt et al., 2018; Brown et al., 2017; Athalye et al., 2017). Many defense methods have also been developed to prevent CNNs from misclassification when facing adversarial attacks. Examples include defensive distillation (Papernot et al., 2016b), training with adversarial examples (Goodfellow et al., 2015), input gradient or curvature regularization (Ross & Doshi-Velez, 2018; Moosavi-Dezfooli et al., 2019), adversarial training via robust optimization (Madry et al., 2018), and TRADES to trade adversarial robustness off against accuracy (Zhang et al., 2019). Besides studying adversarial effects on network prediction decisions, this work explores the connection between adversarial robustness and network interpretability, and provides novel insights on when and how interpretability helps the robustness.

Having a prediction might not be enough for many real-world machine learning applications. It is also crucial to demystify why CNNs make certain decisions. Thus, the problem of network interpretation arises. Various methods have been proposed to understand the mechanism of decision making by CNNs. One category of methods justify a prediction decision by assigning importance values to reflect the influence of individual pixels or image sub-regions on the final classification. Examples include pixel-space sensitivity map methods (Simonyan et al., 2013; Zeiler & Fergus, 2014; Springenberg et al., 2014; Smilkov et al., 2017; Sundararajan et al., 2017) and class-discriminative localization methods (Zhou et al., 2016; Selvaraju et al., 2017; Chattopadhay et al., 2018; Petsiuk et al., 2018), where the former evaluates the sensitivity of a network classification decision to pixel variations at the input, and the latter localizes which parts of an input image were looked at by the network for making a classification decision. Sensitivity map methods include vanilla gradient (Simonyan et al., 2013), deconvolution (Zeiler & Fergus, 2014), guided backpropagation (Springenberg et al., 2014), SmoothGrad (Smilkov et al., 2017), integrated gradient (IG) (Sundararajan et al., 2017) to name a few. They highlight fine-grained details in the image but are not class-discriminative for visual explanation. By contrast, localization approaches like class activation map (CAM) (Zhou et al., 2016), GradCAM (Selvaraju et al., 2017), GradCAM++ (Chattopadhay et al., 2018) and RISE (Petsiuk et al., 2018) are highly class-discriminative, namely, localizing image sub-regions reasoned

for a prediction class. We refer readers to Sec. 2 for some representative interpretation methods. Besides interpreting CNNs via feature importance maps, some methods zoom into the internal response of neural networks. Examples include network dissection (Bau et al., 2017), which evaluates the alignment between individual hidden units and semantic concepts, and learning perceptually-aligned representations from robust training (Engstrom et al., 2019).

Very recently, some works (Xu et al., 2019b;a; Zhang et al., 2018; Subramanya et al., 2018; Ghorbani et al., 2019; Dombrowski et al., 2019; Chen et al., 2019) are beginning to study adversarial robustness by exploring the spectrum between classification accuracy and network interpretability. It was shown in (Xu et al., 2019b;a) that an imperceptible adversarial perturbation to fool classifiers can lead to a significant change in a class-specific network interpretability map, e.g., CAM. Thus, it was argued that such an *interpretability discrepancy* can be used as a helpful metric to differentiate adversarial examples from benign inputs. However, the work (Zhang et al., 2018; Subramanya et al., 2018) showed that under certain conditions, generating an attack (which we call *interpretability sneaking attack, ISA*) that fools the classifier as well as its coupled interpreter (in terms of keeping interpretability map highly similar to that of benign input) is not significantly more difficult than generating adversarial inputs deceiving the classifier only. **Besides investigating robustness in classification through the lens of interpretability, the work (Ghorbani et al., 2019; Dombrowski et al., 2019) studied the robustness in network interpretation maps, showing that which can significantly be manipulated via imperceptible input perturbations but keeping the classifier's decision *intact*. We call this type of threat model *attack against interpretability (AAI)*. The existing work had *no agreement* on the relationship between robustness in interpretation and robustness in classification. Spurred by that, we attempt to explore this relationship from both attack and defense perspectives.**

The most relevant work to ours is (Chen et al., 2019), which robustified network interpretation with the aid of integrated gradient (IG), an axiomatic attribution map. It proposed a robust attribution training, which was shown as a principled generalization of previous formulations of robust classification and an effective defense against AAI. **In this paper, we first investigate when ISA is possible, and then relate our insights on ISA to robust classification and robust interpretability.** Different from the previous work, our paper contains the following contributions.

1. **We provide an answer to the question of when adversarial examples can bypass interpretability discrepancy.** We show that enforcing stealthiness of adversarial examples from network interpretation could be challenging. Its difficulty relies on how one measures the interpretability discrepancy caused by input perturbations.

2. We propose an $\ell_1$ norm based 2-class interpretability discrepancy measure and theoretically show that constraining it helps adversarial robustness.

3. We develop an interpretability-aware robust training method and empirically show that interpretability alone can be used to defend adversarial attacks for both misclassifcation and misinterpretation. Compared to the IG-based robust attribution training (Chen et al., 2019), our approach is simpler in implementation, and provides better robustness even as facing a strong adversary.

## 2 PRELIMINARIES AND MOTIVATION: INTERPRETABILITY OF CNNs FOR JUSTIFYING A CLASSIFICATION DECISION

To explain what and why CNNs predict, we consider two types of network interpretation methods: a) *class activation map (CAM)* (Zhou et al., 2016; Selvaraju et al., 2017; Chattopadhay et al., 2018) and b) *pixel sensitivity map (PSM)* (Simonyan et al., 2013; Springenberg et al., 2014; Smilkov et al., 2017; Sundararajan et al., 2017; Yeh et al., 2019). Let $f(\mathbf{x}) \in \mathbb{R}^C$ denote a CNN-based predictor that maps an input $\mathbf{x} \in \mathbb{R}^d$ to a probability vector of $C$ classes. Here $f_c(\mathbf{x})$, the $c$th element of $f(\mathbf{x})$, denotes the classification score (given by logit before the softmax) for class $c$. Let $L(\mathbf{x}, c)$ denote an interpreter (CAM or PSM) that reflects where in $\mathbf{x}$ contributes to the classifier's decision on $c$.

**CAM-type methods.** CAM (Zhou et al., 2016) produces a class-discriminative localization map for CNNs, which performs global averaging pooling over convolutional feature maps prior to the softmax. Let the penultimate layer output $K$ feature maps, each of which is denoted by a vector

representation $\mathbf{A}_k \in \mathbb{R}^u$ for channel $k \in [K]$. Here $[K]$ represents the integer set $\{1, 2, \ldots, K\}$. The $i$th entry of CAM $L_{\text{CAM}}(\mathbf{x}, c)$ is given by

$$[L_{\text{CAM}}(\mathbf{x}, c)]_i = \frac{1}{u} \sum_{k \in [K]} w_k^c A_{k,i}, \ i \in [u], \tag{1}$$

where $w_k^c$ is the linear classification weight that associates the channel $k$ with the class $c$, and $A_{k,i}$ denotes the $i$th element of $\mathbf{A}_k$. The rationale behind (1) is that the classification score $f_c(\mathbf{x})$ can be written as the average of CAM values (Zhou et al., 2016), $f_c(\mathbf{x}) = \sum_{i=1}^u [L_{\text{CAM}}(\mathbf{x}, c)]_i$. For visual explanation, $L_{\text{CAM}}(\mathbf{x}, c)$ is often up-sampled to the input dimension $d$ using bi-linear interpolation.

GradCAM (Selvaraju et al., 2017) generalizes CAM for CNNs without the architecture 'global average pooling $\rightarrow$ softmax layer' over the final convolutional maps. Specifically, the weight $w_k^c$ in (1) is given by the gradient of the classification score $f_c(\mathbf{x})$ with respect to (w.r.t.) the feature map $\mathbf{A}_k$, $w_k^c = \frac{1}{u} \sum_{i=1}^u \frac{\partial f_c(\mathbf{x})}{\partial A_{k,i}}$. GradCAM++ (Chattopadhay et al., 2018), a generalized formulation of GradCAM, utilizes a more involved weighted average of the (positive) pixel-wise gradients but provides a better localization map if an image contains multiple occurrences of the same class. In this work, we focus on CAM since it is computationally light and our models used in experiments follow the architecture 'global average pooling $\rightarrow$ softmax layer'.

**PSM-type methods.** PSM uses calculations with gradients to assign importance scores to individual pixels toward explaining the classification decision about an input. Examples of commonly-used approaches include vanilla gradient (Simonyan et al., 2013), guided backpropogation (Springenberg et al., 2014), SmoothGrad (Smilkov et al., 2017), and integrated gradient (IG) (Sundararajan et al., 2017). In particular, IG satisfies the *completeness* attribution axiom that PSM ought to obey. Specifically, it averages gradient saliency maps for interpolations between an input $\mathbf{x}$ and a baseline image $\mathbf{a}$:

$$[L_{\text{IG}}(\mathbf{x}, c)]_i = (x_i - a_i) \int_{\alpha=0}^1 \frac{\partial f_c(\mathbf{a} + \alpha(\mathbf{x} - \mathbf{a}))}{\partial x_i} d\alpha \approx (x_i - a_i) \sum_{i=1}^m \frac{\partial f_c(\mathbf{a} + \frac{i}{m}(\mathbf{x} - \mathbf{a}))}{\partial x_i} \frac{1}{m}, \ i \in [d], \tag{2}$$

where $m$ is the number of steps in the Riemman approximation of the integral. The *completeness* axiom (Sundararajan et al., 2017, Proposition 1) states that $\sum_{i=1}^d [L_{\text{IG}}(\mathbf{x}, c)]_i = f_c(\mathbf{x}) - f_c(\mathbf{a})$, where the baseline image $\mathbf{a}$ is often chosen such that $f_c(\mathbf{a}) \approx 0$, e.g., the black image. Note that CAM also satisfies the *completeness* axiom. PSM is able to highlight fine-grained details in the image, but is computationally intensive and not very class-discriminative compared to CAM (Selvaraju et al., 2017).

**Interpretability discrepancy caused by adversarial perturbation.** Let $\mathbf{x}' = \mathbf{x} + \boldsymbol{\delta}$ represent an *adversarial example* w.r.t. $\mathbf{x}$, where $\boldsymbol{\delta}$ denotes an *adversarial perturbation*. By replacing the input image $\mathbf{x}$ with $\mathbf{x}'$, the CNN will be fooled from the *true label* $t$ to the *target (incorrect) label* $t'$. It was recently shown in (Xu et al., 2019b;a) that the adversary could introduce an evident *interpretability discrepancy* w.r.t. *both* the true and the target label in terms of $L(\mathbf{x}, t)$ vs. $L(\mathbf{x}', t)$, and $L(\mathbf{x}, t')$ vs. $L(\mathbf{x}', t')$. An illustrative example is provided in Figure 1. We see that an adversary *suppresses* the network interpretation w.r.t. the true label but *promotes* the interpretation w.r.t. the target label. We also observe that compared to IG, CAM and GradCAM++ better localize class-specific discriminative regions. **These results reveal two observations on how measuring interpretation discrepancy affects classification robustness: a) interpretability discrepancy may be used to detect adversarial examples, b) interpretability discrepancy itself may be vulnerable to adversarial perturbations.** In what follows, we explore the spectrum between adversarial robustness and interpretability from a unified perspective **considering both the adversarial vulnerability of interpretability discrepancy and the value of interpretability discrepancy in a defense** .

## 3  ROBUSTNESS IN CLASSIFICATION VS. INTERPREATION MAP: AN ATTACK PERSPECTIVE

In this section, we examine two types of threat models, *interpretability sneaking attack (ISA)* and *attack against interpretability (AAI)*. Since an adversarial example designed for misclassification

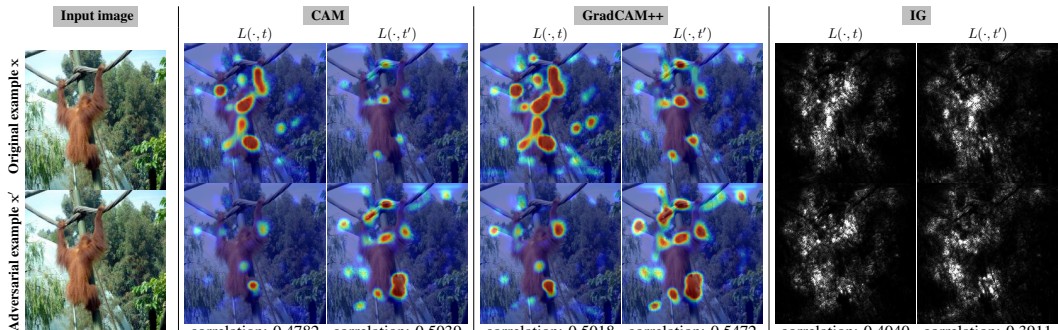

Figure 1: Interpretation ($L$) of benign ($\mathbf{x}$) and adversarial ($\mathbf{x}'$) image from restricted ImageNet (Engstrom et al., 2019) with respect to the true label $t$='monkey' and the target label $t'$='fish'. Here the adversarial example is generated by 10-step PGD attack with perturbation size 0.02 (Madry et al., 2018), and we consider three types of interpretation maps, CAM, GradCAM++ and IG. Given an interpretation method, the first column is $L(\mathbf{x}, t)$ versus $L(\mathbf{x}', t)$, the second column is $L(\mathbf{x}, t')$ versus $L(\mathbf{x}', t')$, and all maps under each category are normalized w.r.t. their largest value. At the bottom of each column, we quantify the resulting interpretability discrepancy by Kendall's Tau order rank correlation (Selvaraju et al., 2017) between every pair of $L(\mathbf{x}, i)$ and $L(\mathbf{x}', i)$ for $i = t$ or $t'$.

gives rise to interpretability discrepancy (which could then be used as a detector for the adversarial input), the problem of ISA arises: One may wonder whether or not it is easy to generate adversarial examples that *mistake classification* but keep *interpretation intact*. **Such adversarial vulnerability could have serious consequences when classification and interpretation are jointly used in tasks like medical diagnosis (Subramanya et al., 2018), and call into question the faithfulness of interpretation to network classification.** One the other hand, it is suggested from interpretability discrepancy that an interpreter could be quite sensitive to input perturbations. Thus, the problem of AAI arises: One may wonder if perturbed inputs could induce *differing explanations* but *without altering predictions*.

## 3.1 RETHINKING ISA FROM PERSPECTIVE OF INTERPRETABILITY DISCREPANCY

Previous work (Zhang et al., 2018; Subramanya et al., 2018) showed that it is *not difficult to hide* adversarial examples from network interpretation when the interpretability discrepancy is measured w.r.t. a *single* class label (either the true label $t$ or the target label $t'$). However, we see from Figure 1 that the adversary (against classification) alters interpretability maps w.r.t. *both* $t$ and $t'$. This motivates us to rethink whether the single-class interpretability discrepancy measure is proper, and whether ISA is truly easy to bypass an interpretability discrepancy check.

We consider the following generic form of interpretability discrepancy

$$\mathcal{D}\left(\mathbf{x}, \mathbf{x}'\right) = \frac{1}{|\mathcal{C}|} \sum_{i \in \mathcal{C}} \left\| L(\mathbf{x}, i) - L(\mathbf{x}', i) \right\|_p, \tag{3}$$

where recall that $\mathbf{x}$ and $\mathbf{x}'$ are natural and adversarial examples respectively, $L$ represents an interpreter, e.g., CAM or IG, $\mathcal{C}$ denotes the set of class labels used in $L$, $|\mathcal{C}|$ is the cardinality of $\mathcal{C}$, and we consider $p \in \{1, 2\}$ in this paper. Clearly, a specification of (3) relies on the choice of $\mathcal{C}$ and $p$. We specify (3) with $\mathcal{C} = \{t, t'\}$ and $p = 1$, leading to $\ell_1$ *2-class interpretability discrepancy measure*,

$$\mathcal{D}_{2,\ell_1}\left(\mathbf{x}, \mathbf{x}'\right) = \frac{1}{2} \left( \left\| L(\mathbf{x}, t) - L(\mathbf{x}', t) \right\|_1 + \left\| L(\mathbf{x}, t') - L(\mathbf{x}', t') \right\|_1 \right). \tag{4}$$

**Rationale behind $\ell_1$ 2-class discrepancy measure.** Compared to the previous work (Zhang et al., 2018; Subramanya et al., 2018) using a single class label, we choose $\mathcal{C} = \{t, t'\}$[1], motivated by the fact that an interpretability discrepancy occurs w.r.t. both $t$ and $t'$ (Figure 1). Moreover, although Euclidean distance (namely, $\ell_2$ norm or its square) is arguably one of the most commonly-used discrepancy metrics (Zhang et al., 2018), we show in Proposition 1 that the proposed interpretability discrepancy measure $\mathcal{D}_{2,\ell_1}\left(\mathbf{x}, \mathbf{x}'\right)$ has a non-trivial lower bound for any successful adversarial attack. This provides an explanation on why it could be difficult to hide adversarial examples from

---

[1]In addition to the *2-class* case, our experiments will also cover the *all-class* case $\mathcal{C} = [C]$.

network interpretability methods. Moreover, $\ell_1$ is an upper bound of the $\ell_2$ norm and promotes the sparsity of interpretability discrepancy, which enforces pixels of $L$ to stay intact when facing input perturbations.

**Proposition 1.** *Given a classifier $f(\mathbf{x}) \in \mathbb{R}^C$ and its interpreter $L(\mathbf{x}, c)$ for $c \in [C]$, suppose that the interpreter satisfies the completeness axiom, namely, $\sum_i [L(\mathbf{x}, c)]_i = f_c(\mathbf{x})$. For a natural example $\mathbf{x}$ and an adversarial example $\mathbf{x}'$ with prediction $t$ and $t'$ respectively, $\mathcal{D}_{2,\ell_1}(\mathbf{x}, \mathbf{x}')$ in (4) has the lower bound*

$$\mathcal{D}_{2,\ell_1}\left(\mathbf{x}, \mathbf{x}'\right) \geq \frac{1}{2}\left(f_t(\mathbf{x}) - f_{t'}(\mathbf{x})\right). \tag{5}$$

**Proof**: See proof in Appendix A. $\qquad\square$

Proposition 1 connects $\mathcal{D}_{2,\ell_1}\left(\mathbf{x}, \mathbf{x}'\right)$ with the classification margin $f_t(\mathbf{x}) - f_{t'}(\mathbf{x})$. Thus, if a classifier has a large classification margin on the natural example $\mathbf{x}$, it will be difficult to find a successful adversarial attack with small interpretability discrepancy. In other words, *constraining the interpretability discrepancy prevents misclassification* since generating adversarial examples becomes infeasible under $\mathcal{D}_{2,\ell_1}\left(\mathbf{x}, \mathbf{x}'\right) < \frac{1}{2}\left(f_t(\mathbf{x}) - f_{t'}(\mathbf{x})\right)$. Also, the completeness condition of $L$ suggests specifying (4) with CAM (1) or IG (2). Here we focus on CAM due to its light computation.

**Design of ISA.** We pose the following optimization problem for design of ISA, which not only fools a classifier's decision but also minimizes the resulting interpretability discrepancy,

$$\begin{aligned}
\underset{\boldsymbol{\delta}}{\text{minimize}} \quad & \lambda \max\{\max_{j \neq t'} f_j(\mathbf{x} + \boldsymbol{\delta}) - f_{t'}(\mathbf{x} + \boldsymbol{\delta}), -\tau\} + \mathcal{D}_{2,\ell_1}\left(\mathbf{x}, \mathbf{x} + \boldsymbol{\delta}\right) \\
\text{subject to} \quad & \|\boldsymbol{\delta}\|_\infty \leq \epsilon,
\end{aligned} \tag{6}$$

where $\lambda > 0$ is a regularization parameter that strikes a balance between the success of an attack and its resulting interpretability discrepancy, $\tau \geq 0$ (e.g., $0.1$ used in the paper) is a tolerance on the classification margin of a successful attack between the target label $t'$ and the non-target top-1 prediction label $\arg\max_{j \neq t'} f_j(\mathbf{x} + \boldsymbol{\delta})$, $\mathcal{D}_{2,\ell_1}$ was defined by (4), and $\epsilon > 0$ is a (pixel-level) perturbation size. In (6), the first term of the objective corresponds to a C&W-type attack loss (Carlini & Wagner, 2017), which reaches $-\tau$ if the attack succeeds in misclassification. To find ISA of *minimum* interpretability discrepancy, we perform a *bisection* on $\lambda$ until there exists no successful attack that can be found when $\lambda$ further decreases. Although we focus on $\ell_\infty$ attack in (6), but a similar formulation applies to $\ell_1$ and $\ell_2$ attacks. Attacks are found using PGD, with subgradients taken at non-differentiable points. We consider only targeted attacks to better evaluate the effect on interpretability of target classes, although this approach can be extended to an untargeted setting (e.g., by using the target label-free interpretability discrepancy measure introduced in the next section).

**Difficulty of generating ISA versus interpretability discrepancy measure.** Through an illustrative example in Figure 2, we elaborate on how the choice of interpretability discrepancy measure plays a crucial role on the difficulty of hiding adversarial examples from network interpretation. We refer readers to Sec. 5 for more experimental results. We generate ISA under different specifications of (3) for different values of perturbation size $\epsilon$. Compared to $\ell_1/\ell_2$ 1-class (true class $t$), $\ell_2$ 2-class, and $\ell_1/\ell_2$ all-class, we see that the $\ell_1$ 2-class interpretability discrepancy (4) cannot be easily mitigated even as the attack power (in terms of $\epsilon$) increases. This is verified by a) its high interpretability discrepancy score and b) its flat slope of discrepancy score against $\epsilon$ in Figure 2-(a)&(b). Figure 2-(c) further shows CAMs of adversarial examples w.r.t. the true label $t$ and the target label $t'$ generated by $\ell_1$ 1/2/all-class ISAs. We note that both 1-class measure and all-class measure could give a *false* sense of ease of hiding adversarial examples. **For $\ell_1$ 1-class ISA, although the interpretability discrepancy w.r.t. $t$ is minimized, the discrepancy w.r.t. $t'$ remains large, supported by the observation that $\ell_1$ 1-class ISA even yields a smaller correlation between $L(\mathbf{x}', t')$ and $L(\mathbf{x}, t')$ than PGD attack. Similarly, although the averaged discrepancy over all classes is minimized for $\ell_1$ all-class ISA, discrepancies w.r.t. specific labels such as $t$ and $t'$ do not necessarily reduce. This illustrates that although ISA may be found, with the proper choice of discrepancy measure, ISA with a low discrepancy becomes quite difficult.**

### 3.2 ATTACK AGAINST INTERPRETABILITY (AAI)

Different from ISA, AAI produces input perturbations to maximize the interpretability discrepancy while keeping the classification decision intact. Thus, AAI provides a means to evaluate the ad-

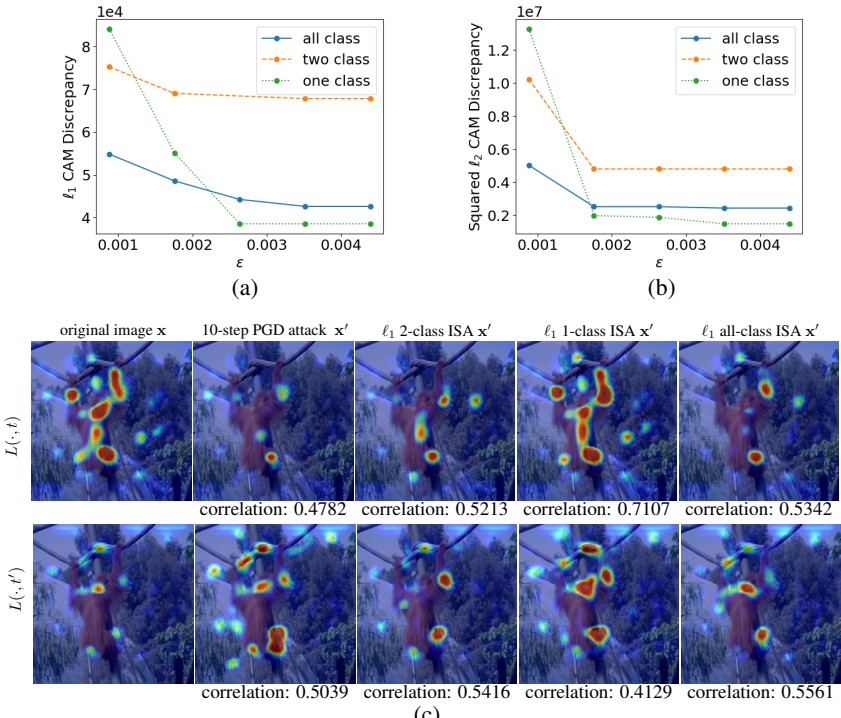

Figure 2: Seeing the effect of discrepancy measure on hiding adversarial examples from network interpretation. The same benign image in Figure 1 is considered. (a) ISA using CAM-based $\ell_1$ 1/2/all-class discrepancy measure versus perturbation size $\epsilon$, (b) ISA using CAM-based squared $\ell_2$ 1/2/all-class discrepancy measure versus $\epsilon$, (c) CAM interpretation of example in Figure 1 and its adversarial counterparts from PGD attack and different specifications of ISA. All interpretation maps are normalized w.r.t. their largest value. At the bottom of each interpretation map $L(\mathbf{x}', \cdot)$, we quantify the interpretability discrepancy by Kendall's Tau order rank correlation between every pair of $L(\mathbf{x}', i)$ and $L(\mathbf{x}, i)$ for $i \in \{t, t'\}$, where $\mathbf{x}'$ is obtained from PGD attack or each specification of ISA.

versarial robustness in interpretability. Since $t = \arg\max_i f_i(\mathbf{x}) = \arg\max_i f_i(\mathbf{x}') = t'$ in AAI, the 2-class interpretability discrepancy measure (4) reduces to its 1-class version. The problem of generating AAI is then cast as

$$\begin{aligned} \underset{\boldsymbol{\delta}}{\text{minimize}} \quad & \lambda \max\{\max_{j \neq t} f_j(\mathbf{x} + \boldsymbol{\delta}) - f_t(\mathbf{x} + \boldsymbol{\delta}), 0\} - \mathcal{D}_1(\mathbf{x}, \mathbf{x} + \boldsymbol{\delta}) \\ \text{subject to} \quad & \|\boldsymbol{\delta}\|_\infty \leq \epsilon, \end{aligned} \quad (7)$$

where the first term is a hinge loss to enforce $f_t(\mathbf{x} + \boldsymbol{\delta}) \geq \max_{j \neq t} f_j(\mathbf{x} + \boldsymbol{\delta})$, namely, $\arg\max_i f_i(\mathbf{x}') = t$ (unchanged prediction under input perturbations), and $\mathcal{D}_1$ denotes a 1-class interpretability discrepancy measure, e.g., $\mathcal{D}_{1,\ell_1}$ from (4), or the top-$k$ pixel difference between interpretability maps (Ghorbani et al., 2019). Similar to (6), the regularization parameter $\lambda$ in (7) strikes a balance between stealthiness in classification and variation in interpretability. Experiments in Sec. 5 will show that the state-of-the-art defense methods against adversarial examples do not necessarily preserve robustness in interpretability as $\epsilon$ increases, although the prediction is not altered.

# 4 INTERPRETABILITY-AWARE ROBUST TRAINING: DEFENSE PERSPECTIVE

We recall from Sec. 3.1 that adversarial examples that intend to fool a classifier could be difficult to evade the $\ell_1$ 2-class interpretability discrepancy. Thus, constraining the interpretability discrepancy helps to prevent misclassification. **Motivated by this observation about ISA, we introduce a interpretability based defense method that penalizes interpretability discrepancy to achieve high classification robustness. As an additional benefit, it also robustifies the classifier against AAI, where the adversary maximizes the interpretability discrepancy.**

**Target label-free interpretability discrepancy.** Different from attack generation, the $\ell_1$ 2-class discrepancy measure (4) cannot directly be used by a defender since the target label $t'$ specified by the adversary is *not* known *a priori*. To circumvent this issue, we propose to approximate the interpretability discrepancy w.r.t. the target label by weighting discrepancies from all non-true classes according to their importance in prediction. This modifies (4) to

$$\tilde{\mathcal{D}}\left(\mathbf{x},\mathbf{x}'\right) = \frac{1}{2}\left\|L(\mathbf{x},t) - L(\mathbf{x}',t)\right\|_1 + \frac{1}{2}\sum_{i \neq t}\frac{e^{f(\mathbf{x}')_i}}{\sum_i e^{f(\mathbf{x}')_i}}||L(\mathbf{x},i) - L(\mathbf{x}',i)||_1, \qquad (8)$$

where the softmax function $\frac{e^{f(\mathbf{x}')_i}}{\sum_i e^{f(\mathbf{x}')_i}}$ adjusts the importance of non-true labels according to their classification confidence. Clearly, when $\mathbf{x}'$ succeeds in misclassification, the top-1 predicted class of $\mathbf{x}'$ becomes the target label and the resulting interpretability discrepancy is most penalized.

**Interpretability-aware robust training.** We propose to train a classifier against the *worst-case* interpretability discrepancy (8), yielding the min-max optimization problem

$$\underset{\boldsymbol{\theta}}{\text{minimize}}\ \mathbb{E}_{(\mathbf{x},t)\sim\mathcal{D}_{\text{train}}}\left[f_{\text{train}}(\boldsymbol{\theta};\mathbf{x},t) + \gamma \underset{\mathbf{x}':\|\mathbf{x}'-\mathbf{x}\|_\infty \leq \epsilon}{\text{maximize}}\ \tilde{\mathcal{D}}\left(\mathbf{x},\mathbf{x}'\right)\right], \qquad (9)$$

where $\boldsymbol{\theta}$ denotes the model parameters, $\mathcal{D}_{\text{train}}$ denotes the training dataset, $f_{\text{train}}$ is the training loss (e.g., cross-entropy loss), $\gamma > 0$ is a regularization parameter, and for ease of notation we omit the parameters $\boldsymbol{\theta}$ and $t$ in $\tilde{\mathcal{D}}\left(\mathbf{x},\mathbf{x}'\right)$. **Here $\gamma$ controls the tradeoff between clean accuracy and robustness; see Appendix C for experiments analyzing this tradeoff.**

In problem (9), the inner maximization is only used to evaluate the worst-case interpretability discrepancy. Thus, it is different from adversarial training (Madry et al., 2018), where the *training loss* is replaced with the *adversarial loss* $\text{maximize}_{\mathbf{x}':\|\mathbf{x}'-\mathbf{x}\|_\infty \leq \epsilon} f_{\text{train}}(\boldsymbol{\theta},\mathbf{x}';\mathbf{x},t)$. The formulation (9) allows us to examine whether or not robust interpretation is directly beneficial to robust classification. For completeness, we will also provide experiment results on a modified formulation of (9) with the use of the adversarial loss.

**Difference to (Chen et al., 2019).** The recent work (Chen et al., 2019) proposed improving adversarial robustness by leveraging robust IG attributions. However, different from (Chen et al., 2019), our approach is motivated by the importance of $\ell_1$ 2-class interpretability discrepancy measure. We will show in Sec. 5 that the incorporation of interpretability discrepancy w.r.t. target class labels, namely, the second term in (8), plays a key role in boosting classification and interpretation robustness. **This is because robust interpretability implies robust classification only when interpretation maps are measured with a proper metric.** We will also show that our proposed method is sufficient to improve adversarial robustness even in the absence of adversarial loss, while the robust attribution regularization method (Chen et al., 2019) becomes ineffective when the attack becomes stronger. Last but not the least, beyond IG, our proposed theory and method apply to any network interpretation method with completeness axiom. The use of CAM avoids Riemman approximation used in IG and thus simplifies the implementation during robust training.

## 5 EXPERIMENTS

In this section, we empirically show the effectiveness of our proposed methods in various attack and defense settings. For ISA, we examine how the interpretability discrepancy measure plays a role in hiding adversarial examples from network interpretation. For AAI, we evaluate its attack success rate under the natural and various robust models. For interpretability-aware robust training, we demonstrate its advantages in a) defending against PGD attacks with different steps and perturbation sizes (Madry et al., 2018; Athalye et al., 2018), b) defending against unforeseen adversarial attacks (Kang et al., 2019), c) rendering robustness in interpretability, and d) computation efficiency compared to the IG-based robust attribution regularization method (Chen et al., 2019).

**Datasets, CNN models, and experiment setting.** We evaluate networks trained on the MNIST, CIFAR-10 and a restricted ImageNet (R-ImageNet) dataset used in (Engstrom et al., 2019). We consider three models, *Small* (for MNIST and CIFAR), *Pool* (for MNIST) and *WResnet* (for CIFAR and R-ImageNet): 1) a small CNN architecture consisting of three convolutional layers of 16, 32

and 100 filters (Small), 2) a CNN architecture with two convolutional layers of 32 and 64 filters each followed by max-pooling which is adapted from (Madry et al., 2018) (Pool), and 3) a Wide Resnet from (Madry et al., 2018) (WResnet). **We refer readers to Appendix B for more details.**

In the following experiments, we consider 5 *baselines* from the literature: i) standard training (*Normal*), ii) adversarial training (*Adv*) (Madry et al., 2018), iii) *TRADES* (Zhang et al., 2019), iv) *IG-Norm* that uses IG-based robust attribution regularization (Chen et al., 2019), v) *IG-Sum-Norm* (namely, IG-Norm with adversarial loss). Additionally, we consider 3 *variants* of our method: vi) the proposed interpretability-aware robust training method (9) (we call *Int*), vii) Int using $\ell_1$ 1-class discrepancy (*Int-1-class*), and viii) Int with adversarial loss (*Int-Adv*). **For interpretability-based methods, the regularization parameter $\gamma$ is set to 0.01. In Appendix C, we explore different settings of the parameter $\gamma$ for our method to demonstrate that we can achieve a range of different points on the robustness-accuracy tradeoff.**

| Dataset | Interpretation method | $\ell_1$ norm | | | $\ell_2$ norm | | |
|---|---|---|---|---|---|---|---|
| | | 1-class | 2-class | all-class | 1-class | 2-class | all-class |
| MNIST | CAM | 3.0723/0.0810 | 3.2672/0.0223 | 2.5289/0.0414 | 0.3061/0.1505 | 0.5654/0.0321 | 0.4320/0.0459 |
| | GradCAM++ | 3.1264/0.0814 | 3.1867/0.0221 | 2.5394/0.0366 | 0.3308/0.1447 | 0.5531/0.0289 | 0.4392/0.0456 |
| | IG | 6.3604/0.0330 | 6.7884/0.0233 | 4.3667/0.2314 | 0.4476/0.0082 | 0.5766/0.0064 | 0.2160/0.0337 |
| | Repr | n/a | 2.3668/0.0404 | n/a | n/a | 0.4129/0.0429 | n/a |
| CIFAR-10 | CAM | 1.9523/0.1450 | 2.5020/0.0496 | 1.7898/0.0774 | 0.1313/0.2369 | 0.3613/0.0668 | 0.2746/0.0809 |
| | GradCAM++ | 1.9355/0.1439 | 2.4788/0.0513 | 1.8020/0.0745 | 0.1375/0.2346 | 0.3577/0.0676 | 0.2758/0.0769 |
| | IG | 4.9499/0.0188 | 4.9794/0.0177 | 2.8541/0.1356 | 0.1230/0.0110 | 0.1309/0.0092 | 0.0878/0.0235 |
| | Repr | n/a | 1.7049/0.0785 | n/a | n/a | 0.1288/0.0056 | n/a |
| R-ImageNet | CAM | 49.286/0.1005 | 61.975/0.0331 | 49.877/0.0557 | 1.9373/0.1526 | 2.6036/0.0791 | 2.0935/0.0863 |
| | GradCAM++ | 39.761/0.1028 | 50.303/0.0453 | 42.390/0.0552 | 1.9185/0.1609 | 2.5869/0.0891 | 2.1151/0.0896 |
| | Repr | n/a | 46.892/0.0657 | n/a | n/a | 2.0730/0.0781 | n/a |

Table 1: NDS and NSL (format given by NDS/NSL) of successful ISAs generated under different specifications of interpretability discrepancy measure (3) and datasets MNIST, CIFAR-10 and R-ImageNet. Here a discrepancy measure with large NDS and small NSL indicates a strong resistance to ISA.

**Evaluating ISA.** We evaluate the effect of interpretability discrepancy measure on ease of finding ISAs. Spurred by Figure 2, such an effect is quantified by calculating minimum discrepancy required in generating ISAs against different values of perturbation size $\epsilon$ in (6). We conduct experiments over 4 network interpretation methods: i) CAM, ii) GradCAM++, iii) IG, and iv) internal representation at the penultimate (pre-softmax) layer (denoted by *Repr*). In order to fairly compare among different interpretation methods, we compute a *normalized discrepancy score (NDS)* extended from (3) and a *normalized slope (NSL)* that measures the relative change of NDS for $\epsilon \in [\check{\epsilon}, \hat{\epsilon}]$. **We refer readers to Appendix D for more details.**

In Table 1, we present NDS and NSL of ISAs generated under different realizations of interpretability discrepancy measure (3), each of which is given by a combination of interpretation method (CAM, GradCAM++, IG or Repr), $\ell_p$ norm ($p \in \{1, 2\}$) and number of interpreted classes. Note that Repr is independent of the number of classes, and thus we report NDS and NSL corresponding to Repr in the 2-class column of Table 1. Given an $\ell_p$ norm and an interpretation method, we consistently find that the use of 2-class measure achieves the largest NDS and smallest NSL at the same time. This implies that the 2-class discrepancy measure increases the difficulty of ISA to evade network interpretability check. Moreover, given a class number and an interpretation method, we see that NDS under $\ell_1$ norm is greater than that under $\ell_2$ norm, since the former is naturally an upper bound of the latter. Also, the use of $\ell_1$ norm often yields a smaller value of NSL, implying that the $\ell_1$-norm based discrepancy measure is more resistant to ISA. Furthermore, by fixing the combination of $\ell_1$ norm and 2 classes, we observe that IG is the most resistant to ISA due to their relatively high NDS and low ISA, and Repr yields the worst performance. However, compared to CAM, the computation cost of IG increases dramatically as the input dimension, the model size, and the number of steps in Riemman approximation increase. We find that it becomes infeasible to generate ISA using IG for WResnet under R-ImageNet within 200 hours.

**Evaluating AAI.** We specify problem (7) under the top-k attack setting (Ghorbani et al., 2017), where the top-k intersection between the original and adversarial interpretability maps is minimized. The strength of AAI is then measured by the Kendall's Tau order rank correlation between the aforementioned two interpretability maps (Chen et al., 2019). The higher the correlation is, the more robust the model is against AAI. Reported rank correlations are averaged over a test set.

**In Table 2, we present the performance of AAI under multiple perturbation sizes to attack models trained using different training methods.** For the model Small under MNIST, we evaluate AAI over 5 baselines (Normal, Adv, TRADES, IG-Norm, IG-Sum-Norm) and 3 variants of our method (Int, Int-1-class and Int-Adv). For a larger model WResnet under CIFAR-10, the training methods using IG (IG-Norm and IG-Norm-Sum) are excluded due to their prohibitive computation cost. The insights learnt from Table 2 are summarized as below. First, the normally trained model (Normal) does not automatically give robustness guarantees in interpretability, particularly for AAI with $\epsilon \geq 0.2$. Second, the methods Adv, TRADES, IG-Sum-Norm and Int-Adv that uses adversarial loss offer certain robustness against AAI but the performance gets worse as the perturbation size $\epsilon$ increases. Third, in the absence of adversarial loss, the baseline IG-Norm becomes less robust as $\epsilon$ increases. By contrast, our proposed method Int is consistently more robust and its advantage becomes more evident as $\epsilon$ increases, **including when $\epsilon$ is increased beyond the value used for training.**

| Method | $\epsilon = 0.05$ | 0.1 | 0.2 | 0.3 | 0.35 | 0.4 |
|---|---|---|---|---|---|---|
| | | | MNIST, Small | | | |
| Normal | 0.907 | 0.797 | 0.366 | -0.085 | -0.085 | -0.085 |
| Adv | **0.978** | **0.955** | **0.910** | **0.857** | 0.467 | 0.136 |
| TRADES | **0.978** | **0.955** | 0.905 | 0.847 | 0.450 | 0.115 |
| IG-Norm | 0.958 | 0.894 | 0.662 | 0.278 | 0.098 | 0.094 |
| IG-Norm-Sum | 0.976 | 0.951 | 0.901 | 0.850 | **0.659** | **0.389** |
| Int | **0.982** | **0.968** | **0.941** | **0.913** | 0.361 | 0.078 |
| Int-Adv | **0.980** | **0.965** | **0.936** | **0.912** | **0.504** | **0.320** |
| Int-1-class | 0.874 | 0.818 | 0.754 | 0.692 | **0.527** | **0.348** |
| | $\epsilon = 2/255$ | 4/255 | 6/255 | 8/255 | 9/255 | 10/255 |
| | | | CIFAR, WResnet | | | |
| Normal | 0.595 | 0.159 | 0.067 | -0.069 | -0.143 | -0.194 |
| Adv | **0.912** | **0.816** | **0.724** | 0.629 | 0.600 | 0.552 |
| TRADES | **0.918** | **0.832** | **0.747** | **0.652** | **0.609** | **0.575** |
| Int | 0.859 | 0.763 | 0.746 | **0.682** | **0.679** | **0.661** |
| Int-Adv | **0.885** | **0.803** | **0.751** | **0.696** | **0.663** | **0.640** |

Table 2: Performance of AAI for different values of perturbation size $\epsilon$ in terms of Kendall's Tau order rank correlation between the original and adversarial interpretability maps. Best robustness results (corresponding to highest correlation values) are highlighted (**1st**, **2nd**, **3rd**) under each column of a dataset-model pair.

**Evaluating interpretability-aware robust training.** We previously showed in Table 2 that interpretability-aware robust training (Int, Int-Adv) often achieves more robust interpretability than state-of-the-art adversarial training methods particularly for large perturbation size. We next provide a thorough evaluation on how interpretability helps adversarial robustness against misclassification.

*Robustness versus efficiency.* **In Figure 3, we present the training time (left y-axis) and the adversarial test accuracy (right y-axis) for different training methods (x-axis), which are ranked in a decreasing order of computation complexity.** Here the adversarial test accuracy (ATA) is measured **using 200-step ($\ell_\infty$-norm) perturbation sizes $\epsilon = 0.3$ and 0.4** on the Small MNIST model (Madry et al., 2018). Note that all AT-type methods (IG-Norm-Sum, Int-Adv, TRADES and Adv) offer robust classification at $\epsilon = 0.3$ with ATA around 80%. **Among non-AT but interpretability promoted defensive schemes (IG-Norm, Int, Int-1-class), we find that only the proposed Int yields competitive ATA, and outperforms all AT-type methods except Int-Adv at $\epsilon = 0.4$. Particularly,**

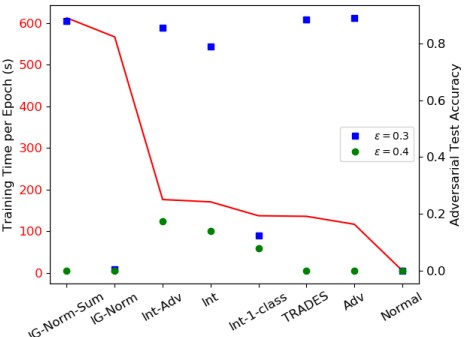

Figure 3: Computation time per epoch and adversarial test accuracy for a Small MNIST model trained with different methods.

**IG-Norm degrades significantly in robust classification when PGD attack becomes stronger (also see results in Table 4).** Moreover, the non-robustness of Int-1-class verifies the importance of 2-class interpretability discrepancy measure on rendering robust classification. Last but not the least, Adv, TRADES and Int-based methods have similar computational complexity, but IG-based methods make training time (per epoch) significantly higher, 3 times more than Int-Adv even under the Small MNIST model.

*Int does **not** cause obfuscated gradients and importance of 2-class measure.* It was shown in (Athalye et al., 2018; Carlini, 2019) that some defense methods cause *obfuscated gradients*, which give a false sense of security. There exist two characteristic behaviors of obfuscated gradients: a) One-step attacks perform better than iterative attacks; b) Increasing distortion bound does not increase success. Motivated by that, we evaluate our interpretability-aware robust training methods under PGD attacks with a) different steps and b) different perturbation sizes. **Table 3 reports ATA of interpretability-aware robust training relative to various baselines, where 200-step PGD attacks are conducted for $\epsilon \in \{0, 0.05, 0.1, 0.2, 0.3, 0.35, 0.4\}$ for MNIST and $\epsilon \in \{0, 2/255, 4/255, 6/255, 8/255, 9/255, 10/255\}$ for CIFAR. We use $\epsilon = 0.3$ on MNIST**

**and** $8/255$ **on CIFAR for robust training.** As we can see, ATA decreases as $\epsilon$ increases. Thus, our methods do not exhibit the behavior a) of obfuscated gradients. **We also observe that compared to Adv and TRADES, Int achieves reasonable but worse ATA as** $\epsilon < 0.3$ **on MNIST and** $\epsilon \leq 8/255$ **on CIFAR. As** $\epsilon$ **used in PGD attack achieves the value used for robust training, Int achieves ATA** $0.790$ **against ATA** $0.890$ **from Adv on MNIST, but outperforms Adv on CIFAR** ($0.270$ **vs** $0.170$**). Interestingly, the advantage of Int becomes more evident as the adversary becomes stronger, i.e.,** $\epsilon > 0.3$ **on MNIST and** $\epsilon > 8/255$ **on CIFAR.** We highlight that such a robust classification is achieved by promoting robustness of interpretability alone (without using adversarial loss). It is worth mentioning that IG-Norm fails to defend PGD attack with $\epsilon = 0.3$ for the Small MNIST model. We further note that Int-1-class performs much worse than Int, supporting the *importance of using 2-class* discrepancy measure (see Prop. 1). Besides robust classification, IG-Norm and Int-1-class are also not sufficient to render robustness in interpretation (Table 2).

Table 4 shows ATA of interpretability-aware robust training against $k$-step PGD attacks (examining behavior b) of obfuscated gradients), where $k \in \{1, 10, 100, 200\}$. As we can see, ATA decreases as $k$ increases. This also suggests that the high robust accuracy from our methods is not a result of obfuscated gradients. Similar to Table 3, we see that compared to Int, IG-Norm and Int-1-class are insufficient to defend PGD attacks with $k \geq 100$.

*Beyond $\ell_\infty$-norm PGD attacks.* In Table 5, we present ATA of interpretability-aware robust training and various baselines for defending attacks (Gabor, Snow, JPEG $\ell_\infty$, JPEG $\ell_2$, and JPEG $\ell_1$) recently proposed in (Kang et al., 2019). These attacks are called 'unforseen attacks' since they are not met by PGD-based robust training and often induce larger perturbations than conventional PGD attacks. For robust training methods without resorting to adversarial loss, we find that Int significantly outperforms IG-Norm especially under Snow and JPEG $\ell_p$ attacks. Int also yields quite competitive robustness compared to AT-type methods on most attacks.

| Method | $\epsilon = 0$ | 0.05 | 0.1 | 0.2 | 0.3 | 0.35 | 0.4 |
|---|---|---|---|---|---|---|---|
| | | | | MNIST, Small | | | |
| Normal | **1.000** | 0.530 | 0.045 | 0.000 | 0.000 | 0.000 | 0.000 |
| Adv | **0.980** | **0.960** | **0.940** | **0.925** | **0.890** | 0.010 | 0.000 |
| TRADES | 0.970 | **0.970** | **0.955** | **0.930** | **0.885** | 0.000 | 0.000 |
| IG-Norm | **0.985** | **0.950** | 0.895 | 0.410 | 0.005 | 0.000 | 0.000 |
| IG-Norm-Sum | 0.975 | 0.955 | 0.935 | **0.910** | 0.880 | **0.115** | 0.000 |
| Int-1-class | 0.975 | 0.635 | 0.330 | 0.140 | 0.125 | **0.115** | **0.080** |
| Int | 0.935 | 0.930 | 0.905 | 0.840 | 0.790 | **0.180** | **0.140** |
| Int-Adv | 0.950 | 0.945 | 0.905 | 0.880 | **0.855** | **0.355** | **0.175** |
| | $\epsilon = 0$ | 2/255 | 4/255 | 6/255 | 8/255 | 9/255 | 10/255 |
| | | | | CIFAR, WResnet | | | |
| Normal | **0.765** | 0.250 | 0.070 | 0.060 | 0.060 | 0.060 | 0.060 |
| Adv | **0.720** | **0.605** | **0.485** | **0.330** | **0.170** | **0.145** | 0.085 |
| TRADES | **0.765** | **0.610** | **0.460** | 0.295 | **0.170** | 0.140 | **0.100** |
| Int | **0.735** | **0.630** | **0.485** | **0.365** | **0.270** | **0.240** | **0.210** |
| Int-Adv | 0.665 | 0.585 | **0.510** | **0.385** | **0.320** | **0.300** | **0.280** |
| Int-1-class | 0.685 | 0.505 | 0.360 | 0.190 | 0.065 | 0.040 | 0.025 |

Table 3: 200-step PGD accuracy for different values of perturbation size $\epsilon$. Best ATA results are highlighted (**1st**, **2nd**, **3rd**) at each column. Note that ATA with $\epsilon = 0$ reduces to natural accuracy.

| Method | Steps= 1 | 10 | 100 | 200 |
|---|---|---|---|---|
| | MNIST, Small, $\epsilon = 0.3$ | | | |
| Normal | **0.990** | 0.070 | 0.000 | 0.000 |
| Adv | **0.975** | **0.945** | **0.890** | **0.890** |
| TRADES | **0.970** | **0.955** | **0.885** | **0.885** |
| IG-Norm | **0.970** | 0.905 | 0.005 | 0.005 |
| IG-Norm-Sum | 0.970 | **0.940** | **0.880** | **0.880** |
| Int-1-class | 0.950 | 0.365 | 0.125 | 0.125 |
| Int | 0.935 | 0.910 | 0.790 | 0.790 |
| Int-Adv | 0.950 | 0.905 | 0.855 | 0.855 |
| | CIFAR, Wresnet, $\epsilon = 8/255$ | | | |
| Normal | 0.470 | 0.075 | 0.060 | 0.060 |
| Adv | **0.590** | **0.205** | **0.185** | **0.185** |
| TRADES | **0.590** | 0.180 | 0.165 | 0.165 |
| Int | **0.620** | **0.310** | **0.275** | **0.275** |
| Int-Adv | **0.580** | **0.345** | **0.335** | **0.335** |
| Int-1-class | 0.505 | 0.100 | 0.060 | 0.060 |

Table 4: Multi-step PGD accuracy. Best ATA results are highlighted (**1st**, **2nd**, **3rd**) at each column.

# 6 CONCLUSION

In this paper, we investigate the connection between network interpretability and adversarial robustness. We show theoretically and empirically that with the correct choice of discrepancy measure, it is difficult to hide adversarial examples from interpretation. We leverage this discrepancy measure to develop a interpretability-aware robust training method that displays 1) high classification robustness in a variety of settings and 2) high robustness of interpretation.

| Method | Gabor | Snow | JPEG $\ell_\infty$ | JPEG $\ell_2$ | JPEG $\ell_1$ |
|---|---|---|---|---|---|
| | | CIFAR-10, Small | | | |
| Normal | 0.125 | 0.000 | 0.000 | 0.030 | 0.000 |
| Adv | **0.190** | **0.115** | **0.460** | **0.380** | **0.230** |
| TRADES | **0.220** | 0.085 | 0.425 | 0.300 | 0.070 |
| IG-Norm | 0.155 | 0.015 | 0.000 | 0.000 | 0.000 |
| IG-Norm-Sum | **0.185** | **0.110** | **0.480** | **0.375** | 0.215 |
| Int | 0.160 | 0.105 | **0.440** | **0.345** | **0.260** |
| Int-Adv | 0.150 | **0.120** | 0.340 | 0.310 | **0.235** |

Table 5: ATA on different unforseen attacks in (Kang et al., 2019). Best results in each column are highlighted (**1st**, **2nd**, **3rd**)

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

## APPENDIX

## A PROOF OF PROPOSITION 1

For $\forall c \in [C]$, by the completeness axiom we have $f_c(\mathbf{x}) = \sum_i [L(\mathbf{x}, c)]_i$. Using this fact, we obtain that

$$
\begin{aligned}
f_{t'}(\mathbf{x}') - f_{t'}(\mathbf{x}) &= \sum_i ([L(\mathbf{x}', t')]_i - [L(\mathbf{x}, t')]_i) \\
&\leq \sum_i |[L(\mathbf{x}', t')]_i - [L(\mathbf{x}, t')]_i| = \|L(\mathbf{x}', t') - L(\mathbf{x}, t')\|_1.
\end{aligned}
\tag{10}
$$

Similarly, we have

$$
f_t(\mathbf{x}) - f_t(\mathbf{x}') \leq \|L(\mathbf{x}, t) - L(\mathbf{x}', t)\|_1.
\tag{11}
$$

Adding (10) and (11) rearranging yields

$$
[f_{t'}(\mathbf{x}') - f_t(\mathbf{x}')] + [f_t(\mathbf{x}) - f_{t'}(\mathbf{x})] \leq \|L(\mathbf{x}', t') - L(\mathbf{x}, t')\|_1 + \|L(\mathbf{x}, t) - L(\mathbf{x}', t)\|_1.
\tag{12}
$$

Since $f_{t'}(\mathbf{x}') - f_t(\mathbf{x}') \geq 0$, we then have $\|L(\mathbf{x}', t') - L(\mathbf{x}, t')\|_1 + \|L(\mathbf{x}, t) - L(\mathbf{x}', t)\|_1 \geq f_t(\mathbf{x}) - f_{t'}(\mathbf{x})$, which is equivalent to (5).

# B   ADDITIONAL DETAILS ON MODEL AND DATASET

**The considered network model has a global average pooling layer followed by a fully connected layer at the end of the network. For R-ImageNet, we use only a normally trained network, used for evaluating ISA. During adversarial training of all baselines, 40 adversarial steps are used for MNIST and 10 steps for CIFAR. To ensure stability of all training methods, the size of perturbation is increased during training from $0$ to a final value of $0.3$ on MNIST and $8/255$ on CIFAR. MNIST and R-ImageNet networks are trained for 100 epochs and CIFAR networks are trained for 200 epochs. A batch size of 50 is used for MNIST, 128 for CIFAR and 64 for R-Imagenet. For all methods, training is performed using Adam with an initial learning rate of 0.0001 for MNIST and 0.001 for CIFAR and R-Imagenet, with the learning decayed by $\times 1/10$ at training steps 40000 and 60000 for CIFAR and 8000 and 16000 for R-Imagenet.**

# C EXPERIMENTS ON REGULARIZATION PARAMETER $\gamma$

**We conduct experiments for evaluating the sensitivity of the regularization parameter $\gamma$ in our proposed apporoach (namely, Int) under a Small MNIST model. Adversarial test accuracy (ATA) and clean test accuracy results are plotted in Figure A1. As illustrated, using different values of the hyperparameter $\gamma$ controls the tradeoff between clean accuracy and ATA, with smaller $\gamma$ yielding higher clean accuracy, but lower ATA (a value of $\gamma = 0$ corresponds to normal training). We note that with the model tested, ATA stops increasing at a value of $\gamma = 0.01$. Beyond this value, clean accuracy continues to decrease while ATA slightly decreases. These results indicate that by choosing an appropriate $\gamma$, it is possible to smoothly interpolate between normal training and maximally robust Int training.**

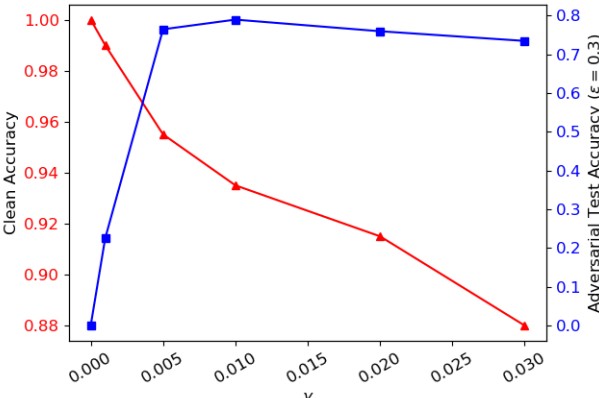

Figure A1: Clean test accuracy and adversarial test accuracy for a Small MNIST model trained with Int using different values of regularization parameter $\gamma$.

# D    DETAILS ON NDS AND NSL

**In order to fairly compare among different interpretation methods, we compute a *normalized discrepancy score (NDS)* extended from (3):** $\mathcal{D}_{\text{norm}} = \frac{1}{|\mathcal{C}|} \sum_{i \in \mathcal{C}} \left\| \frac{\mathcal{L}(\mathbf{x}, i) - \mathcal{L}(\mathbf{x}', i)}{\max_j L(\mathbf{x}, i)_j - \min_j L(\mathbf{x}, i)_j} \right\|_p$. **A smaller value of NDS implies the less difficulty for ISA to hide adversarial examples. A strong ISA is also expected to resist interpretability discrepancy even as the perturbation size $\epsilon$ increases; see Figure 2-(a)&(b). To quantify this, we compute an additional quantity called *normalized slope (NSL)* that measures the relative change of NDS for $\epsilon \in [\check{\epsilon}, \hat{\epsilon}]$:** $\mathcal{S}_{\text{norm}} = \frac{|\mathcal{D}_{\text{norm}}^{(\hat{\epsilon})} - \mathcal{D}_{\text{norm}}^{(\check{\epsilon})}| / \mathcal{D}_{\text{norm}}^{(\check{\epsilon})}}{(\hat{\epsilon} - \check{\epsilon})/\check{\epsilon}}$. **The smaller NDS is, the less strong ISA is to evade network interpretation change. In our experiment, we choose $\check{\epsilon} = \epsilon^*$ and $\hat{\epsilon} = 1.6\,\epsilon^*$, where $\epsilon^*$ is the minimum perturbation size required for a successful PGD attack for each image x. Here we perform binary search over $\epsilon$ to find its smallest value for misclassification. Reported NDS and NSL statistics are averaged over a test set.**

# E ADDITIONAL TABLES

| Method | $\epsilon = 0$ | 0.05 | 0.1 | 0.2 | 0.3 |
|---|---|---|---|---|---|
| MNIST, Pool | | | | | |
| Normal | 0.990 | 0.435 | 0.070 | 0.000 | 0.000 |
| Adv | 0.930 | 0.885 | 0.835 | 0.695 | 0.535 |
| TRADES | 0.955 | 0.910 | 0.870 | 0.720 | 0.455 |
| IG-Norm | 0.980 | 0.940 | 0.660 | 0.050 | 0.000 |
| IG-Norm-Sum | 0.920 | 0.885 | 0.840 | 0.700 | 0.540 |
| Int | 0.950 | 0.930 | 0.875 | 0.680 | 0.390 |
| Int-Adv | 0.870 | 0.840 | 0.810 | 0.755 | 0.690 |
| Int-1-class | 0.975 | 0.885 | 0.720 | 0.200 | 0.130 |
| | $\epsilon = 0$ | 2/255 | 4/255 | 6/255 | 8/255 |
| CIFAR-10, Small | | | | | |
| Normal | 0.650 | 0.015 | 0.000 | 0.000 | 0.000 |
| Adv | 0.505 | 0.470 | 0.380 | 0.330 | 0.285 |
| TRADES | 0.630 | 0.465 | 0.355 | 0.235 | 0.140 |
| IG-Norm | 0.525 | 0.435 | 0.360 | 0.295 | 0.230 |
| IG-Norm-Sum | 0.390 | 0.365 | 0.325 | 0.310 | 0.285 |
| Int | 0.530 | 0.450 | 0.345 | 0.290 | 0.215 |
| Int-Adv | 0.675 | 0.145 | 0.005 | 0.000 | 0.000 |
| Int-1-class | 0.515 | 0.450 | 0.380 | 0.315 | 0.265 |

Table A1: 200 steps PGD accuracy, additional results.

| Method | $\epsilon = 0$ | 0.05 | 0.1 | 0.2 | 0.3 |
|---|---|---|---|---|---|
| MNIST, Pool | | | | | |
| Normal | 0.990 | 0.435 | 0.070 | 0.000 | 0.000 |
| Adv | 0.930 | 0.885 | 0.835 | 0.695 | 0.535 |
| TRADES | 0.955 | 0.910 | 0.870 | 0.720 | 0.460 |
| IG-Norm | 0.980 | 0.945 | 0.660 | 0.060 | 0.000 |
| IG-Norm-Sum | 0.920 | 0.885 | 0.840 | 0.700 | 0.540 |
| Int | 0.950 | 0.930 | 0.875 | 0.680 | 0.385 |
| Int-Adv | 0.870 | 0.840 | 0.810 | 0.755 | 0.700 |
| Int-1-class | 0.975 | 0.885 | 0.720 | 0.200 | 0.130 |
| | $\epsilon = 0$ | 2/255 | 4/255 | 6/255 | 8/255 |
| CIFAR-10, Small | | | | | |
| Normal | 0.650 | 0.015 | 0.000 | 0.000 | 0.000 |
| Adv | 0.505 | 0.470 | 0.380 | 0.330 | 0.285 |
| TRADES | 0.630 | 0.465 | 0.355 | 0.235 | 0.140 |
| IG-Norm | 0.525 | 0.435 | 0.360 | 0.295 | 0.230 |
| IG-Norm-Sum | 0.390 | 0.365 | 0.325 | 0.310 | 0.285 |
| Int | 0.530 | 0.450 | 0.345 | 0.290 | 0.215 |
| Int-Adv | 0.675 | 0.145 | 0.005 | 0.000 | 0.000 |
| Int-1-class | 0.515 | 0.450 | 0.380 | 0.315 | 0.265 |

Table A2: 100 step PGD accuracy, additional results.

| Method | $\epsilon = 0$ | 0.05 | 0.1 | 0.2 | 0.3 |
|---|---|---|---|---|---|
| | | MNIST, Pool | | | |
| Normal | 0.990 | 0.470 | 0.135 | 0.135 | 0.135 |
| Adv | 0.930 | 0.885 | 0.845 | 0.845 | 0.845 |
| TRADES | 0.955 | 0.910 | 0.870 | 0.870 | 0.870 |
| IG-Norm | 0.980 | 0.945 | 0.705 | 0.705 | 0.705 |
| IG-Norm-Sum | 0.920 | 0.885 | 0.850 | 0.850 | 0.850 |
| Int | 0.950 | 0.930 | 0.885 | 0.885 | 0.885 |
| Int-Adv | 0.870 | 0.840 | 0.810 | 0.810 | 0.810 |
| Int-1-class | 0.975 | 0.885 | 0.750 | 0.750 | 0.750 |
| | $\epsilon = 0$ | 2/255 | 4/255 | 6/255 | 8/255 |
| | | CIFAR-10, Small | | | |
| Normal | 0.650 | 0.015 | 0.000 | 0.000 | 0.000 |
| Adv | 0.505 | 0.470 | 0.380 | 0.325 | 0.280 |
| TRADES | 0.630 | 0.465 | 0.360 | 0.240 | 0.145 |
| IG-Norm | 0.675 | 0.145 | 0.005 | 0.000 | 0.000 |
| IG-Norm-Sum | 0.515 | 0.450 | 0.380 | 0.315 | 0.265 |
| Int | 0.525 | 0.435 | 0.360 | 0.295 | 0.235 |
| Int-Adv | 0.390 | 0.365 | 0.325 | 0.310 | 0.285 |
| Int-1-class | 0.530 | 0.450 | 0.345 | 0.290 | 0.220 |

Table A3: 10 step PGD accuracy, additional results.

| Method | $\epsilon = 0.05$ | 0.1 | 0.2 | 0.3 |
|---|---|---|---|---|
| | | MNIST, Pool | | |
| Normal | 0.934 | 0.876 | 0.719 | 0.482 |
| Adv | 0.976 | 0.951 | 0.896 | 0.824 |
| TRADES | 0.976 | 0.952 | 0.891 | 0.815 |
| IG-Norm | 0.942 | 0.872 | 0.648 | 0.341 |
| IG-Norm-Sum | 0.976 | 0.951 | 0.895 | 0.824 |
| Int | 0.964 | 0.928 | 0.852 | 0.771 |
| Int-Adv | 0.977 | 0.957 | 0.921 | 0.891 |
| Int-1-class | 0.930 | 0.871 | 0.779 | 0.704 |
| | $\epsilon = 2/255$ | 4/255 | 6/255 | 8/255 |
| | | CIFAR-10, Small | | |
| Normal | 0.694 | 0.350 | 0.116 | -0.031 |
| Adv | 0.958 | 0.907 | 0.849 | 0.783 |
| TRADES | 0.940 | 0.867 | 0.781 | 0.689 |
| IG-Norm | 0.810 | 0.552 | 0.308 | 0.131 |
| IG-Norm-Sum | 0.958 | 0.907 | 0.847 | 0.779 |
| Int | 0.965 | 0.926 | 0.883 | 0.840 |
| Int-Adv | 0.979 | 0.956 | 0.931 | 0.904 |
| Int-1-class | 0.961 | 0.918 | 0.871 | 0.820 |

Table A4: Kendall rank correlation coefficients of top-k CAM attacks against interpretability found using 200 steps of PGD, additional results.

| Method | $\epsilon = 0.05$ | 0.1 | 0.2 | 0.3 |
|---|---|---|---|---|
| | MNIST, Pool | | | |
| Normal | 0.957 | 0.912 | 0.811 | 0.594 |
| Adv | 0.978 | 0.955 | 0.903 | 0.839 |
| TRADES | 0.978 | 0.955 | 0.900 | 0.822 |
| IG-Norm | 0.979 | 0.955 | 0.879 | 0.771 |
| IG-Norm-Sum | 0.979 | 0.956 | 0.905 | 0.843 |
| Int | 0.949 | 0.893 | 0.759 | 0.588 |
| Int-Adv | 0.983 | 0.967 | 0.938 | 0.909 |
| Int-1-class | 0.934 | 0.858 | 0.747 | 0.717 |
| | $\epsilon = 2/255$ | 4/255 | 6/255 | 8/255 |
| | CIFAR-10, Small | | | |
| Normal | 0.691 | 0.335 | 0.118 | -0.031 |
| Adv | 0.958 | 0.908 | 0.850 | 0.781 |
| TRADES | 0.939 | 0.866 | 0.779 | 0.690 |
| IG-Norm | 0.805 | 0.546 | 0.303 | 0.117 |
| IG-Norm-Sum | 0.957 | 0.906 | 0.844 | 0.773 |
| Int | 0.964 | 0.925 | 0.881 | 0.831 |
| Int-Adv | 0.975 | 0.949 | 0.919 | 0.886 |
| Int-1-class | 0.959 | 0.914 | 0.861 | 0.803 |

Table A5: Kendall rank correlation coefficients of top-k GradCAM++ attacks against interpretability found using 200 steps of PGD, additional results.

