# OpenReview forum: "Visual Interpretability Alone Helps Adversarial Robustness"
_ICLR.cc/2020/Conference — Reject_

### Official Review · AnonReviewer2 · 2019-10-24
**Official Blind Review #2**

**Rating:** 6

**Review:**

Interesting work and good contribution
#Summary:
The paper demonstrated that by having an l1-norm based 2-class interpretability discrepancy measure, it can be shown both empirically and theoretically that it is actually difficult to hide adversarial examples. Furthermore, the authors propose an interpretability-aware robust training method and show it can be used to successfully defend adversarially attacks and can result in comparable performance compared to adversarial training.

#Strength
The paper is well written and structured, with a clear demonstration of technical details. Compared with other works that tried to use model interpretation to help improve the model’s robustness, the authors not only consider the saliency map computed for the actual target label but also the label that corresponds to the adversarial example. The proposed interpretability discrepancy measure is novel and has been proven effective to defend interpretability sneaking attacks that aiming to fool both classifiers and detectors and against interpretability-only attacks. Furthermore, extensive experiments have been done to prove the effectiveness of interpretability-aware training, which strengthens the claims of the entire paper.

#Presentation
Good coverage of the literature in both adversarial robustness and model interpretation.
Some minor typos need to be fixed. For example, in the second last line of the caption of Figure. 2, one L(x’,i) should be L(x,i) if I understand correctly.

**Experience Assessment:**

I have read many papers in this area.

**Review Assessment: Checking Correctness Of Derivations And Theory:**

I carefully checked the derivations and theory.

**Review Assessment: Checking Correctness Of Experiments:**

I carefully checked the experiments.

**Review Assessment: Thoroughness In Paper Reading:**

I read the paper thoroughly.

---

> ### Author Response · Authors · 2019-11-11
> **Response to Reviewer #2**
>
> We thank reviewer #2 very much for their positive comments and for accurately summarizing our key contributions. We are very glad to learn that the reviewer found the topic interesting and our study meaningful. Thanks for carefully checking our presentation. We have fixed the typo in Figure 2: Yes, the first $\mathbf x^\prime$ should be $\mathbf x$. We have also carefully  revised our paper for a better version.

---

### Official Review · AnonReviewer3 · 2019-10-24
**Official Blind Review #3**

**Rating:** 3

**Review:**

The present work considers adversarial attacks that also yield similar outputs for "interpretability methods", which are methods that output some vector corresponding to a given classification (usually the vector is e.g. an image or a similar object). It also shows that by regularizing nearby inputs to have similar interpetations (instead of similar classifications), robustness can be achieved similar to adversarial training.

I did not understand the motivation of the paper. Why is it important for adversarial attacks to yield similar interpretations? A human would need to assess the interpretations to detect the attack, but it would already be trivial for an attack to be detected given human oversight (just check whether the classification of the image matches the human-assigned label). It also wasn't clear how this was related to the other observation that regularizing based on interpretability yields robustness; these seem like two fairly separate results.

Finally, I found the claim that "interpretability alone helps robustness" to be misleading and not substantiated by the paper. The purported justification is that regularizing nearby inputs to have the same interpretation yields robustness. But a better summary of this observation is that "robustness of interpretability implies robustness of classification", which is not surprising, and is in fact a trivial corollary of the fact that the metric on interpretations dominates the classification error metric (an observation which is made in the paper).

More minor, but I found it hard to follow the writing in the paper (this is related to the motivation being unclear). This is exacerbated by the paper being longer than unusual (10 pages instead of 8).

**Experience Assessment:**

I have published in this field for several years.

**Review Assessment: Checking Correctness Of Derivations And Theory:**

I assessed the sensibility of the derivations and theory.

**Review Assessment: Checking Correctness Of Experiments:**

I did not assess the experiments.

**Review Assessment: Thoroughness In Paper Reading:**

I made a quick assessment of this paper.

---

> ### Author Response · Authors · 2019-11-11
> **Response to Reviewer #3 (Part 1)**
>
> We thank reviewer #3 for the valuable comments.
>
> #General response on the motivation of our work#
>
> In what follows, we would like to further clarify the motivation of our work. The manuscript is updated to make our points clearer.
>
> The primary motivation of our work is to investigate the relationship between network interpretability and adversarial robustness. Based on previous literature, there are seemingly two possible hypotheses for how robust interpretability affects adversarial robustness of classification: (a) From an attack perspective (Zhang et al., 2018; Subramanya et al., 2018), robust interpretability does not significantly help robust classification since empirically adversarial examples with a small interpretation discrepancy may have a large classification error. (b) From a defense perspective (Chen et al., 2019), robust interpretability helps robust classification since robust interpretations make aspects of the network less sensitive to small perturbations.
>
> We aim to make a unified answer to the debate between (a) and (b). We found that the choice of interpretability discrepancy matters when drawing conclusions. We showed that with an appropriate choice of interpretability discrepancy (namely, the proposed $\ell_1$ norm based 2-class measure), the claim (a) may NOT be correct since hiding adversarial examples from network interpretation could be difficult (see Prop. 1 and the corresponding detailed examples and analysis in Sec. 3.1). Moreover, our interpretability-aware robust training results provide a positive answer to (b): robust interpretability does help robust classification by solely penalizing the proposed interpretability discrepancy (see Sec. 4) during training but the choice of interpretability discrepancy again matters.
>
> References:
>
> Xinyang Zhang, et al., Interpretable deep learning under fire.
> Akshayvarun Subramanya, et al., Towards hiding adversarial examples from network interpretation. arXiv preprint arXiv:1812.02843, 2018.
> J. Chen, et al.,Robust attribution regularization, NeurIPS, 2019.
>
>
>
> # Question: “Why is it important for adversarial attacks to yield similar interpretations?” #
>
>
> Response: Adversarial attacks that yield similar interpretations is a recent threat model (we call interpretability sneaking attack (ISA)) proposed by (Zhang et al., 2018; Subramanya et al., 2018), which showed that there exist adversarially chosen perturbations that fool an image classifier but minimize discrepancy of corresponding interpretation maps. If these perturbations are possible, then it could create confusion between the model interpreter and the classifier, and it could further confuse AI systems which use network interpretations in down-stream actions, e.g., transfer learning (Shafahi et al. 2019) and medical recommendation (G Quellec, et al.). Thus, we believe that ISA is a practical attacking scenario to study the relationship between network interpretation and classification.
>
> However, the goal of this paper is not to make a claim that ISA is a strong attack. Instead, we would like to have a deeper understanding on the plausibility of ISA. We found that ISA can fail when the 2-class interpretability discrepancy was examined; see Figure 2 and more results in experiments. This showed that hiding adversarial attack from network interpretation is challenging. Its difficulty relies on how one measures the interpretability discrepancy caused by input perturbations.
>
> G Quellec, et al. Deep image mining for diabetic retinopathy screening. Medical image analysis, 39:178, 2017.
> Shafahi, Ali, et al. "Adversarially robust transfer learning." arXiv preprint arXiv:1905.08232 (2019).

---

> > ### Author Response · Authors · 2019-11-11
> > **Response to Reviewer #3 (Part 2)**
> >
> > # Question: Is a human necessary for interpretability-driven defenses to adversarial examples? #
> >
> >
> > Response: In our work, including others on adversarial examples and network interpretation (Chen et al., 2019), a human is not required to inspect the interpretation to make a defensive decision. First, the information on the human assigned label is usually not accessed to detect the adversarial example, since the adversarial example is a testing-phase threat model (Metzen et al., 2017), and the human is not assumed to be in the loop when DL/ML models make inferences. Second, we would like to highlight that our work is not based on detection of adversarial attacks. Even for interpretability-based detectors of adversarial attacks (Tao et al., 2018), a human is not required.  Indeed, if a human inspection was available, then a human could assign a label herself even in the absence of ML/DL classification. Third, in our proposed interpretability-aware robust training method, a human is not needed to inspect the interpretation. Instead of relying on interpretations to be viewed from a human perspective, our defense uses a 2-class interpretability metric itself as a robustness measure.  By regularizing with the proposed interpretability discrepancy penalty in the standard training objective, the learnt model yields robust classification against testing-phase adversarial examples; see its comparable performance to state-of-the-art adversarial training method (using adversarial loss) in Figure 3.
> >
> > Jan Hendrik Metzen, Tim Genewein, Volker Fischer, Bastian Bischoff. On Detecting Adversarial Perturbations. ICLR 2017.
> > Guanhong Tao, Shiqing Ma, Yingqi Liu, Xiangyu Zhang. Attacks Meet Interpretability: Attribute-steered Detection of Adversarial Samples. NeurIPS 2018.
> > Jiefeng Chen, Xi Wu, Vaibhav Rastogi, Yingyu Liang, and Somesh Jha. Robust attribution regularization, NeurIPS, 2019.
> >
> >
> >
> >
> >
> > #Question: How is robustness in interpretation related to the observation that regularizing based on interpretability yields robustness? These seem like two fairly separate results.#
> >
> > Response: The observation that regularizing based on proper interpretability metrics yields robustness is used to enforce robustness in classification through the lens of interpretation map discrepancy. The rationale behind that is that constraining the $\ell_1$-norm based two-class interpretability discrepancy helps to prevent misclassification. This fact is both supported theoretically (see Proposition 1) and in our experiments: Our method achieves comparable robust accuracy to adversarial training to standard PGD attacks (27% vs. 17% on CIFAR at $\epsilon=8/255$, 79% vs 89% on MNIST at $\epsilon=0.3$, and 21% vs. 8.5% on CIFAR at $\epsilon=10/255$, 14% vs 0% on MNIST at $\epsilon=0.4$; see Table 3) and unforeseen attacks (44% vs 46% against JPEG $\ell_\infty$, 26% vs 23% against JPEG $\ell_1$ on CIFAR; see Table 5). Here we use $\epsilon = 0.3$ on MNIST and $8/255$ on CIFAR for robust training. The above results show that our approach is able to provide a better robustness even as facing a strong adversary.
> >
> >
> > As another benefit, regularizing based on interpretability yields robustness in interpretation since the interpretability discrepancy is minimized against input perturbations during training. This is supported by the defensive results against attack against interpretability (AAI). Note that AAI is not a threat model belonging to adversarial attacks, since it only manipulates network interpretation maps but keeps classifier’s decision intact. Thus compared to standard defenses such as adversarial training, our defense achieves higher robustness to AAI than adversarial training in terms of rank-correlation of the original and attacked interpretations (0.913 vs 0.857 on MNIST at $\epsilon=0.3$, 0.682 vs 0.629 on CIFAR at $\epsilon=8/255$; see Table 2). In our revision, we have added additional experiments on larger eps values beyond the trained one to additionally illustrate the higher robustness on more powerful AAI (0.349 vs 0.136 on MNIST at $\epsilon=0.4$, 0.661 vs 0.552 on CIFAR at $\epsilon=10/255$; see Table 2).

---

> > > ### Author Response · Authors · 2019-11-11
> > > **Response to Reviewer #3 (Part 3)**
> > >
> > > #Questions: Comments regarding the surprisingness of interpretation robustness implying classification robustness.#
> > >
> > > Response: We thank the reviewer very much on this constructive comment.
> > >
> > > We are sorry to learn that the current title is misleading, and we thank the reviewer’s summary on "robustness of interpretability implies robustness of classification". However, this does not fully align the research theme of the entire paper.  Our detailed response is provided as below.
> > >
> > > In a general setting, robustness of interpretability does NOT alway IMPLY robustness of classification. If the proper interpretability discrepancy metric is not used, we may get negative results. For example, the Int-1-class method and IG-Norm that uses IG-based robust attribution regularization (Chen et al., 2019), which imposes robust interpretation only for the true class of training data, does not result in strong robustness in classification (Int-1-class: 12.5% vs Int(-2-class): 79% on MNIST at $\epsilon=0.3$, Int-1-class: 6.5% vs Int(-2-class): 27% on CIFAR at $\epsilon=8/255$; see Table 3). This also implies that we need careful analysis and studies to achieve this seemingly non-surprising result.
> > >
> > > We also do not think that our study is trivial. To support the result that robustness of interpretability implies robustness of classification, we find that it requires 1) interpretation methods to satisfy the completeness axiom, and 2) the proper choice of $\ell_p$ norm and number of classes in measuring interpretability discrepancy. The point 1) enables us to consider the more computationally-light interpretation method e.g., CAM instead of IG used in (Chen et al., 2019), given the fact  that IG satisfies completeness axiom but is computationally intensive. The point 2) provides a guideline on how to achieve robustness in classification through the lens of interpretability. We have showed that even in the absence of adversarial classification loss, analyzing a proper interpretability discrepancy alone helps adversarial robustness in classification.
> > >
> > > Based on the aforementioned analysis and the concern raised by the reviewer, we would like to highlight the role of the proper interpretability discrepancy in the title, e.g., the new one “Proper Network Interpretability Helps Adversarial Robustness in Classification”.
> > >
> > >
> > > #Question: Comments on paper clarity.#
> > >
> > > Response: We have improved the clarity of our motivations in the revised version. Hopefully, our detailed response and the revised manuscript could address most of the reviewers’ concerns about writing.

---

> > ### Comment · AnonReviewer3 · 2019-11-12
> > **Still confused on motivation**
> >
> > Hello,
> >
> > Thank you for your response. However, I am still confused about the motivation:
> >
> > > Why is it important for adversarial attacks to yield similar interpretations?
> >
> > In your response you cite other papers that consider this threat model, but the fact that other people have considered it does not make it a well-motivated model. I would be most convinced by a self-contained explanation of a scenario where an attacker would need to fool an interpretability method (and not just the classifier) in order for it to be successful.

---

> > > ### Author Response · Authors · 2019-11-12
> > > **Clarification on motivation**
> > >
> > > Thanks for your question.
> > >
> > > The attack (fooling an interpretability method, not just the classifier) could be a threat model for a neural network whose usage relies jointly on interpretation maps and classification results.
> > >
> > > One can use an interpretability method as a beneficial post-hoc supplement to visualize/interpret the prediction result, e.g., localizing the most discriminative image region of a medical image (with respect to the predicted label) might help doctors in disease diagnosis and medical recommendation (Ghorbani et al.; Subramanya et al.). In this case, if the interpreter can be arbitrarily controlled and fooled by the adversary, then it creates inconsistency between an interpreter and a classifier by making the interpretation not reflect the network's classification.  In general, the ISA attack could provide a way to evaluate the consistency between a classifier and an interpreter (different classification vs. similar interpretation). If the inconsistency holds, it can undermine the trustworthiness and transparency of the DL system.
> > >
> > > In addition to specific examples, exploring the relationship between robustness in interpretation and robustness in classification is an interesting scientific question by itself. Previous work has not provided a complete answer on the question of the interpretability-classification relationship in the context of adversarial robustness. Our work, we believe, made an effort and showed that with a proper discrepancy metric, interpretability can not easily be fooled arbitrarily by the adversary when facing input perturbation.
> > >
> > > Amirata Ghorbani, Abubakar Abid, and James Zou. Interpretation of neural networks is fragile. In Proceedings of the AAAI Conference on Artificial Intelligence, volume 33, pp. 3681–3688, 2019.
> > > Akshayvarun Subramanya, et al., Towards hiding adversarial examples from network interpretation. arXiv preprint arXiv:1812.02843, 2018.

---

### Official Review · AnonReviewer1 · 2019-11-08
**Official Blind Review #1**

**Rating:** 3

**Review:**

In summary, this paper studies if interpretation robustness (i.e., similar examples should have similar interpretation) can help enhance the robustness of the model, especially in terms of adversarial attacks. The study direction itself is interesting and very useful for the interpretation and adversarial attack community. Moreover, some promising results can be observed in part of the empirical study. However, this paper can be improved a lot as follows.

1. This paper states several times that "adversarial examples can be hidden from neural network interpretability". It is not clear on the definition of "hidden" in terms of  "interpretability". Therefore, how this "hidden" is related and why this "hidden" is important are unclear too.

2. Many details are missing, which makes the proposal suspicious. For example, the proposed method has a tradeoff parameter \lambda. However, the settings and affects are not discussed at all. Without a clear setup, the reproducibility and applicability is in doubt.

3. Some empirical results are overstated. For example, why 0.790 vs 0.890 and 0.270 vs 0.170  are comparable results? These results show the weakness of the proposed method. Further explanations can be provided. From the reported results, it could be useful to see results when the perturbation is even higher to check the limitation of the proposed method.

4. Besides the clarification in the writing mentioned above, some typos or errors should be fixed, e.g., f_t'(x') - - f_t(x') >=0 in the proof of proposition 1.

**Experience Assessment:**

I have published one or two papers in this area.

**Review Assessment: Checking Correctness Of Derivations And Theory:**

I carefully checked the derivations and theory.

**Review Assessment: Checking Correctness Of Experiments:**

I carefully checked the experiments.

**Review Assessment: Thoroughness In Paper Reading:**

I read the paper thoroughly.

---

> ### Author Response · Authors · 2019-11-11
> **Response to Reviewer #1 (Part 1)**
>
> #Question: 1. This paper states several times that "adversarial examples can be hidden from neural network interpretability". It is not clear on the definition of "hidden" in terms of  "interpretability". Therefore, how this "hidden" is related and why this "hidden" is important are unclear too.#
>
> Response: The definition of “hidden” was borrowed from (Zhang et al., 2018; Subramanya et al., 2018), which refers to the interpretation map of an adversarial example generated from  Interpretability Sneaking Attack (ISA) being visually similar to the original interpretation map of the benign example. We quantify this “hidden” effect with the Kendall’s Tau order rank correlation between interpretation maps before and after performing ISA. As a visual example,
> Figure 2c shows that the conventional ISA generated under 1-class interpretability discrepancy (namely, $\ell_1$ 1-class ISA) minimizes the interpretability discrepancy  w.r.t. the true label $t$ only, supported by a high correlation value 0.7107.
>
> If ISA is a real threat, then it could create confusion between the model interpreter and the classifier, and it could further confuse AI systems which use network interpretations in down-stream actions, e.g., medical recommendation (G Quellec, et al., 2017) and transfer learning (Shafahi et al., 2019). Thus, we think that studying the “hidden” effect of ISA is an important aspect to explore the relationship between network interpretability and robust classification.
>
> However,  the main theme of this paper is not to make a claim that adversarial examples are able to be hidden from network interpretability checkers. Instead, we would like to have a deeper understanding on the plausibility of ISA. We found that ISA can fail when the 2-class interpretability discrepancy was examined; e.g., Figure 2c column 3 versus column 1 (compared to column 4 versus column 1). Our main point for ISA is that hiding adversarial attack from network interpretation is actually challenging. Its difficulty relies on how one measures the interpretability discrepancy caused by input perturbations.
>
>
>
> Xinyang Zhang, et al., Interpretable deep learning under fire.
> Akshayvarun Subramanya, et al., Towards hiding adversarial examples from network interpretation. arXiv preprint arXiv:1812.02843, 2018.
> G Quellec, et al. Deep image mining for diabetic retinopathy screening. Medical image analysis, 39:178, 2017.
> Shafahi, Ali, et al. "Adversarially robust transfer learning." arXiv preprint arXiv:1905.08232 (2019).
>
> #Question: 2. Many details are missing, which makes the proposal suspicious. For example, the proposed method has a tradeoff parameter \lambda. However, the settings and effects are not discussed at all. Without a clear setup, the reproducibility and applicability is in doubt.#
>
> Response:
>
> We apologize for the missing details such that you felt our proposal 'suspicious’.
> In Sec. 5 and Appendix B-D, we have tried our best to present a clear experiment setup. The parameter $\lambda$ used during finding ISA controls the tradeoff between an attack changing the network classification and minimizing the interpretability discrepancy. As explained in section 3.1, $\lambda$ is chosen using the bisection method to find a successful attack with the minimum interpretability discrepancy.
>
> And we have conducted additional experiments on the training regularization parameter $\gamma$. As we can see in Appendix C, Figure A1, $\gamma$ controls the tradeoff between clean accuracy and adversarial test accuracy. The value of $\gamma$ chosen for our experiments ($\gamma=0.01$) is the value yielding the highest robust accuracy on the model tested. Our results indicate that $\gamma$ can be chosen to smoothly interpolate between normal training and maximally robust interpretability-aware training.
>
> We have also released our code for reviewer’s reproducibility and applicability check.

---

> > ### Author Response · Authors · 2019-11-11
> > **Response to Reviewer #1 (Part 2)**
> >
> > #Question: 3. Some empirical results are overstated. For example, why 0.790 vs 0.890 and 0.270 vs 0.170  are comparable results? These results show the weakness of the proposed method. Further explanations can be provided. From the reported results, it could be useful to see results when the perturbation is even higher to check the limitations of the proposed method.#
> >
> > Response:
> > We will make our statement as accurate as possible. We would like to convey that our proposal achieved  robust classification by promoting robustness of interpretability alone (without using the adversarial loss). Compared to Adv and TRADES, our approach Int (2-class without using adversarial loss) achieves reasonable but worse adversarial test accuracy (ATA) as $\epsilon < 0.3$ on MNIST and $\epsilon \leq 8/255$ on CIFAR. As $\epsilon$ used in PGD attack achieves the value (0.3 on MNIST and 8/255 on CIFAR) used for robust training, Int yields ATA $0.790$ against ATA $0.890$ from Adv on MNIST, but outperforms the conventional adversarial training (Adv) on CIFAR ($0.270$ vs $0.170$). Interestingly, the advantage of Int becomes more evident as the adversary becomes stronger, i.e., $\epsilon > 0.3$ on MNIST and $\epsilon > 8/255$ on CIFAR. For example, the newly conducted experiments in Table 3 show that Int outperforms Adv in terms of ATA: 21% vs. 8.5% on CIFAR at eps=10/255, and 14% vs 0% on MNIST at eps=0.4.  We have further provided the details of our experiment results in the revision to make our statement as clear as possible.
> >
> > On the other hand, when comparing to Int-1-class and IG-Norm that use 1-class based interpretability discrepancy measure, our approach generally outperforms them in ATA by large margins (Int: 0.790, Int-1-class: 0.125, IG-Norm: 0.005 on MNIST at $\epsilon = 0.3$, Int: 0.270, Int-1-class: 0.065 on CIFAR at $\epsilon = 8/255$; see Table 3). In this sense, the proposed interpretability-aware robust training methods yields promising results due the appropriate interpretability regularization.
> >
> > Following the reviewer’s suggestion, when the perturbation radius $\epsilon$ becomes much higher, we see more evident benefits of our proposed Int (2-class) approach. On both MNIST and CIFAR models, our approach outperforms adversarial training on $\epsilon$ beyond the one used for training (Int: 0.175, Adv: 0.000 on MNIST at $\epsilon = 0.4$, Int: 0.210, Adv: 0.085 on CIFAR at $\epsilon = 10/255$; see Table 3).
> >
> > #Question: 4. Besides the clarification in the writing mentioned above, some typos or errors should be fixed, e.g., f_t'(x') - - f_t(x') >=0 in the proof of proposition#
> >
> > Response: Thanks for pointing out this typo. We have tried our best to make clear and accurate presentation.

---

### Author Response · Authors · 2019-11-11
**General response to all reviewers and summary of revisions**

We thank all reviewers for their insightful and valuable comments. Our paper has been greatly improved based on these comments. The major modifications are summarized below.

We have updated Sec. 1$\&$2 for a clearer motivation on our research to explore the relationship between network interpretability and adversarial robustness in classification.
We have updated Sec. 3 for elaborating on why we care about interpretability sneaking attack (ISA), how to evaluate the hidden effect, and why interpretability discrepancy matters when generating and evaluating ISA.
We have updated Sec. 4 to reveal a better connection between robustness in interpretation and robustness in classification.
We have updated Sec. 5 and Appendix to clarify our experiment setup, and conducted new experiments on the regularization parameter $\gamma$ as well as the large perturbation radius $\epsilon$; see updated Figure 3, new Figure A1 and updated Tables 2,3 for larger perturbation radius.

We marked our major modifications in BLUE.

---

### Decision · Program_Chairs · 2019-12-19

**Decision:**

Reject

**Comment:**

This work focuses on how one can design models with robustness of interpretations. While this is an interesting direction, the paper would benefit from a more careful treatment of its technical claims.